# Nucleosome destabilization by nuclear non-coding RNAs

Risa Fujita[1,7], Tatsuro Yamamoto[2,3,7], Yasuhiro Arimura[1], Saori Fujiwara[2], Hiroaki Tachiwana[3], Yuichi Ichikawa[3], Yuka Sakata [3], Liying Yang[4], Reo Maruyama[4], Michiaki Hamada[5,6], Mitsuyoshi Nakao[2], Noriko Saitoh[3]* & Hitoshi Kurumizaka [1]*

In the nucleus, genomic DNA is wrapped around histone octamers to form nucleosomes. In principle, nucleosomes are substantial barriers to transcriptional activities. Nuclear non-coding RNAs (ncRNAs) are proposed to function in chromatin conformation modulation and transcriptional regulation. However, it remains unclear how ncRNAs affect the nucleosome structure. Eleanors are clusters of ncRNAs that accumulate around the estrogen receptor-α (*ESR1*) gene locus in long-term estrogen deprivation (LTED) breast cancer cells, and markedly enhance the transcription of the *ESR1* gene. Here we detected nucleosome depletion around the transcription site of Eleanor2, the most highly expressed Eleanor in the LTED cells. We found that the purified Eleanor2 RNA fragment drastically destabilized the nucleosome in vitro. This activity was also exerted by other ncRNAs, but not by poly(U) RNA or DNA. The RNA-mediated nucleosome destabilization may be a common feature among natural nuclear RNAs, and may function in transcription regulation in chromatin.

[1] Laboratory of Chromatin Structure and Function, Institute for Quantitative Biosciences, The University of Tokyo, 1-1-1 Yayoi, Bunkyo-ku, Tokyo 113-0032, Japan. [2] Department of Medical Cell Biology, Institute of Molecular Embryology and Genetics, Kumamoto University, 2-2-1 Honjo, Chuo-ku, Kumamoto 860-0811, Japan. [3] Division of Cancer Biology, The Cancer Institute of Japanese Foundation for Cancer Research, 3-8-31 Ariake, Koto-ku, Tokyo 135-8550, Japan. [4] Project for Cancer Epigenomics, The Cancer Institute of Japanese Foundation for Cancer Research, 3-8-31 Ariake, Koto-ku, Tokyo 135-8550, Japan. [5] Graduate School of Advanced Science and Engineering, Waseda University, 3-4-1 Okubo, Shinjuku-ku, Tokyo 169-8555, Japan. [6] AIST-Waseda University Computational Bio Big-Data Open Innovation Laboratory (CBBD-OIL), 3-4-1 Okubo, Shinjuku-ku, Tokyo 169-8555, Japan. [7] These authors contributed equally: Risa Fujita, Tatsuro Yamamoto. *email: noriko.saitoh@jfcr.or.jp; kurumizaka@iam.u-tokyo.ac.jp

In eukaryotes, genomic DNA is tightly compacted in chromatin[1]. The basic architecture of chromatin is the nucleosome, in which two H2A–H2B dimers and one H3–H4 tetramer form the histone octamer, and about 150 base pairs of DNA are left-handedly wrapped around it[2]. The nucleosome structure is extremely stable, and thus it generally suppresses genomic DNA functions, such as transcription[3–5].

Noncoding RNAs (ncRNAs), which are not translated into proteins, exist from bacteria to mammals, and play pivotal roles in genome regulation, such as transcription and recombination, in eukaryotes[6–13]. In humans, ~75% of the genomic DNA is transcribed[14,15]. However, 98–99% of the RNA transcripts are not translated, and remain as ncRNAs[16]. ncRNAs are transcribed from everywhere in the genome, including intergenic regions, introns, and gene bodies, regardless of the sense and antisense directions[17,18].

The ESR1 gene, encoding the estrogen receptor-α, ER, is upregulated in ER-positive breast cancer cells when they undergo adaptation to the hormone-depleted environment, defined as long-term estrogen deprivation (LTED)[19–24]. A cluster of ncRNAs named Eleanors (ESR1 locus enhancing and activating noncoding RNAs) is highly transcribed from a 0.7-Mb genomic region, including the ESR1 gene, on human chromosome 6. In the LTED cell nucleus, Eleanors accumulate around their own coding sites in cis[25]. Consequently, the transcription of the ESR1 gene is activated in the LTED cells[25]. Similar to Eleanors, other ncRNAs also accumulate in the nucleus[26,27]. However, the molecular mechanism by which nuclear ncRNAs such as Eleanors activate transcription has not been elucidated yet.

Previous formaldehyde-assisted isolation of regulatory elements (FAIRE) data, coupled with massively parallel sequencing data, implied that nucleosomes are remarkably depleted in the upstream region of the ESR1 gene in the LTED cells[24,25,28], but the underlying mechanism remained unknown. In this study, we found that the purified Eleanor2 RNA, which is the most abundant member of the Eleanors, drastically destabilized the nucleosome in vitro, probably by destabilizing the H2A–H2B association with the nucleosome. Other ncRNAs, MALAT1, DSCAM-AS1, and XIST also promoted nucleosome destabilization in vitro. Consistently, depletion of the Eleanor2 RNA reduced chromatin accessibility at its cognate sites in vivo. Therefore, the RNA-mediated nucleosome instability, which could play an important role in transcription regulation, may be a common function among the chromatin-associated nuclear ncRNAs. These novel intrinsic properties of ncRNAs may explain the higher chromatin accessibility, by which Eleanor2 accumulates and activates the nearby gene, ESR1.

## Results

### Nucleosome-free regions are formed in the *Eleanor2* locus.
We previously reported that a cluster of ncRNAs, named Eleanors, is transcribed from a 0.7-Mb region containing the ESR1 gene (chr6:152083078-152424447, hg19) in LTED cells derived from human breast cancer MCF7 cells[25]. To understand the chromatin status in this region in the cells, we investigated the published FAIRE-seq database, which is commonly used to identify nucleosome-free regions in chromatin[24,28]. The FAIRE-seq data (Fig. 1a, b) revealed that nucleosomes may be destabilized in the region upstream of the ESR1 gene in LTED cells, but not in MCF10A, non-tumorigenic mammary epithelial cells. An alignment of the FAIRE-seq and our total RNA-seq data[25] revealed that these possible nucleosome-destabilized regions overlapped with the most abundantly expressed Eleanor peak, which coincided with a previously annotated BRCAT32 ncRNA (A site, red highlight in Fig. 1a). Intensive transcriptome analyses identified BRCAT32 as a breast cancer-associated transcript, but its molecular characterization has not been performed[29,30]. BRCAT32 (chr6:151913411-151937977 (−), hg19), which is immediately upstream of the ESR1 gene, is transcribed from intronic regions of the CCDC170 gene, but in the reverse direction. A 656-base sequence (chr6:151937260-151937915 (−), hg19) in BRCAT32 is highly conserved among mammals, according to both the Phast-Cons and PhyloP methods (http://compgen.cshl.edu/phast/), suggesting that it may be an important RNA unit, and thus we named this RNA fragment Eleanor2 (Fig. 1b).

By qRT-PCR, we confirmed that the Eleanor2 RNA was abundantly produced in LTED cells, but not in MCF10A cells (Fig. 1c). We then performed a fluorescence in situ hybridization (FISH) analysis with the Eleanor2-BAC probe (Fig. 1a, green bar), which detects the Eleanor2 RNA sequence. Fluorescent nuclear foci, representing Eleanor clouds, were detected in LTED cells, but not in MCF10A cells (Fig. 1d). We further performed dual FISH experiments using probes (ESR1-BAC and ESR1-BAC2) that detect the previously reported Eleanors transcribed from within and immediately upstream of the ESR1 gene[25]. We found that an Eleanor2 RNA (green signals in Fig. 1e) accumulated in the proximity of the other Eleanors (red signals in Fig. 1e, also see ref. [25]) in the nucleus. The Eleanor RNA detected by the ESR1-BAC2 probe reportedly associates with its cognate genome locus[25]. In contrast, the transcripts from the ERBB2 gene on chromosome 17, which was activated in LTED cells[25], were not colocalized with the Eleanor2 RNA in LTED cells (Fig. 1e). The Eleanor2 RNA specifically associates with the upstream region of the ESR1 gene, together with other Eleanor RNAs, in LTED cells.

We then tested whether nucleosome destabilization occurs in the ESR1 upstream region, as observed in the FAIRE-seq data (Fig. 1a). To do so, we performed FAIRE-qPCR with the A, B, and C sites, corresponding to the Eleanor2 RNA coding (A site), its neighboring (B and C sites) regions, and the control D site (Fig. 1a). Sites A–C were suggested to be nucleosome-depleted sites by the previous FAIRE-seq analysis with the LTED cells (Fig. 1a). Intriguingly, in the A, B, and C sites, nucleosome depletion was significantly enhanced in the LTED cells, as compared with the non-tumorigenic MCF10A cells (Fig. 1f). At the control D site, nucleosome depletion was not detected by FAIRE-seq in both the LTED and MCF10A cells (Fig. 1a). We confirmed that the nucleosome depletion was not enhanced at the D site in the LTED cells, by a FAIRE-qPCR analysis (Fig. 1f). Collectively, these data implied that the Eleanor2 RNA may actually destabilize the nucleosomes around the upstream region of the ESR1 gene in LTED cells.

**Eleanor2 RNA destabilizes the nucleosome in vitro.** We then tested whether the Eleanor2 RNA affects the nucleosome stability in vitro. The Eleanor2 RNA was produced by in vitro transcription, and was purified to near homogeneity (Supplementary Fig. 1a). The nucleosome was reconstituted with recombinant human histones H2A, H2B, H3.1, and H4, in the presence of 146 base pairs of palindromic alpha-satellite DNA[2], by the salt-dialysis method[31] (Supplementary Fig. 1b, c). We then performed a thermal stability assay (Fig. 2a). In this assay, the thermal dissociation of histones from the nucleosome is independently detected as a biphasic denaturation curve, in which the first phase (65–75 °C) and the second phase (82–86 °C) correspond to the H2A–H2B dissociation and the H3–H4 dissociation, respectively[32]. In the presence of a stoichiometric amount of Eleanor2 RNA (nucleosome: RNA = 1: 1 molar ratio), the first phase for the H2A–H2B dissociation was drastically shifted toward lower temperatures (52–67 °C), although the second phase was

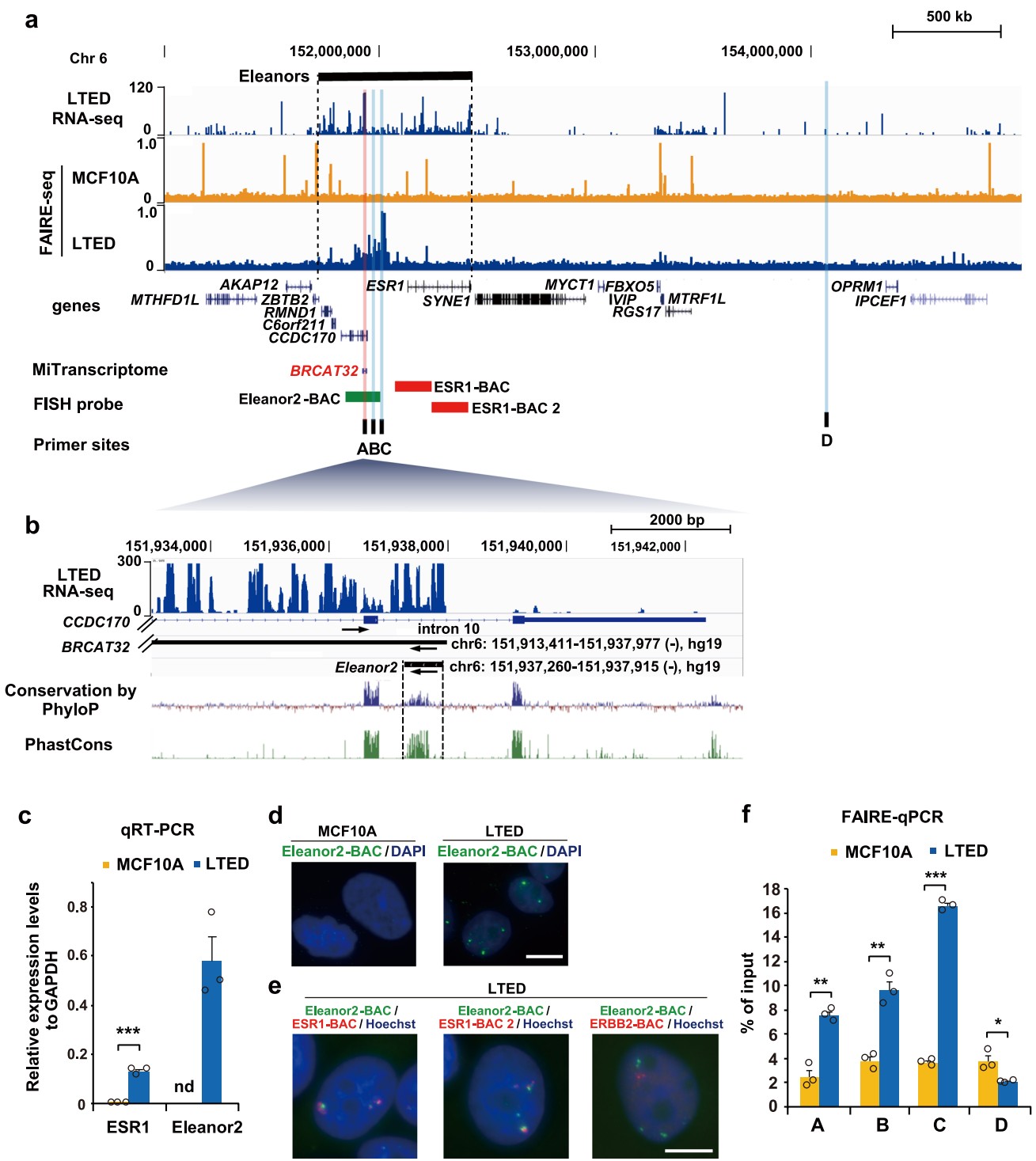

minimally affected (Fig. 2b). In fact, an H3–H4 and DNA complex, in which the DNA is wrapped around the H3–H4 complex without the H2A–H2B dimers, was not destabilized in the presence of a stoichiometric amount of Eleanor2 RNA (Fig. 2c). The H2A–H2B dissociation was not drastically enhanced in the presence of the DNA encoding Eleanor2 (Fig. 2b). These results indicated that the Eleanor2 RNA destabilizes the nucleosome by weakening the H2A–H2B association, without reducing the H3–H4 affinity to the DNA. The H2A–H2B dissociation remained substantially enhanced with one-tenth of the Eleanor2

RNA (nucleosome: RNA = 1: 0.1), and was saturated with one-half of the Eleanor2 RNA (nucleosome: RNA = 1: 0.5) (Fig. 2d).

To determine whether another type of nucleosome positioning sequence has a different sensitivity to the ncRNA, we performed a thermal stability assay using the nucleosome with 147 base pairs of the MMTV (mouse mammary tumor virus)-A DNA sequence[33] (Supplementary Fig. 1d, e). The destabilization profile of the MMTV nucleosome in the presence of the Eleanor2 RNA was similar to that of the alpha-satellite nucleosome (Supplementary Fig. 1f; Fig. 2b). This suggests that nucleosome

**Fig. 1 Eleanor2 RNA promotes an open chromatin conformation in LTED cells. a** Overview of the *Eleanor2* locus and its neighboring region (Chr6:148,180,000–156,290,000). The RNA-seq[25] and FAIRE-seq[24,28] tracks in the indicated cells are aligned against the genome reference GRCh37/hg19. Positions of the UCSC genes and lncRNAs annotated in MiTranscriptome[30]. The BAC DNA used as the FISH probe (green bar: Eleanor2-BAC, red bars: ESR1-BAC and ESR1-BAC2), and the primers used for FAIRE-qPCR (black bars; A–D) in Fig. 1 **d–f** are also shown. The red vertical line indicates the most highly transcribed region of the Eleanors, named Eleanor2, which corresponds to part of the breast cancer-associated lncRNA BRCAT32. The primer site A is in the *Eleanor2*-coding site. Sites A, B, and C were suggested to have open chromatin conformations, while site D was predicted to be closed, according to the FAIRE-seq track above. **b** Enlarged view of the region surrounding *Eleanor2*, which is highly conserved among mammals. Eleanor2 and BRCAT32 are transcribed in the opposite orientation from the *CCDC170* gene (arrows). **c** Eleanor2 RNA is highly expressed in LTED cells. The qRT-PCR values of ESR1 mRNA and Eleanor2 RNA relative to the control GAPDH mRNA are shown. The expression of Eleanor2 RNA was not detectable in MCF10A cells (marked as nd). **d** RNA-FISH visualizing RNA foci containing Eleanor2. The Eleanor2-BAC DNA was used as the probe (green). DNA was counterstained with DAPI (blue). Scale bar, 10 μm. **e** Transcripts from the *Eleanor2* region, but not the *ERBB2* region, were colocalized with the *ESR1* region in the RNA clouds. The BAC-DNA clones were used as the probe. Scale bar, 10 μm. The maximum intensity *z*-projection of each channel is shown. **f** FAIRE-qPCR showing that the Eleanor chromatin forms an open configuration in LTED cells. Values represent amounts of DNA in the nucleosome-free fraction relative to the input DNA. Data presented in **c** and **f** are means ± s.e.m. (*n* = 3, biologically independent samples). *P*-values were calculated using the unpaired, two-tailed, Student's *t* test (\**P* < 0.05, \*\**P* < 0.01, \*\*\**P* < 0.001).

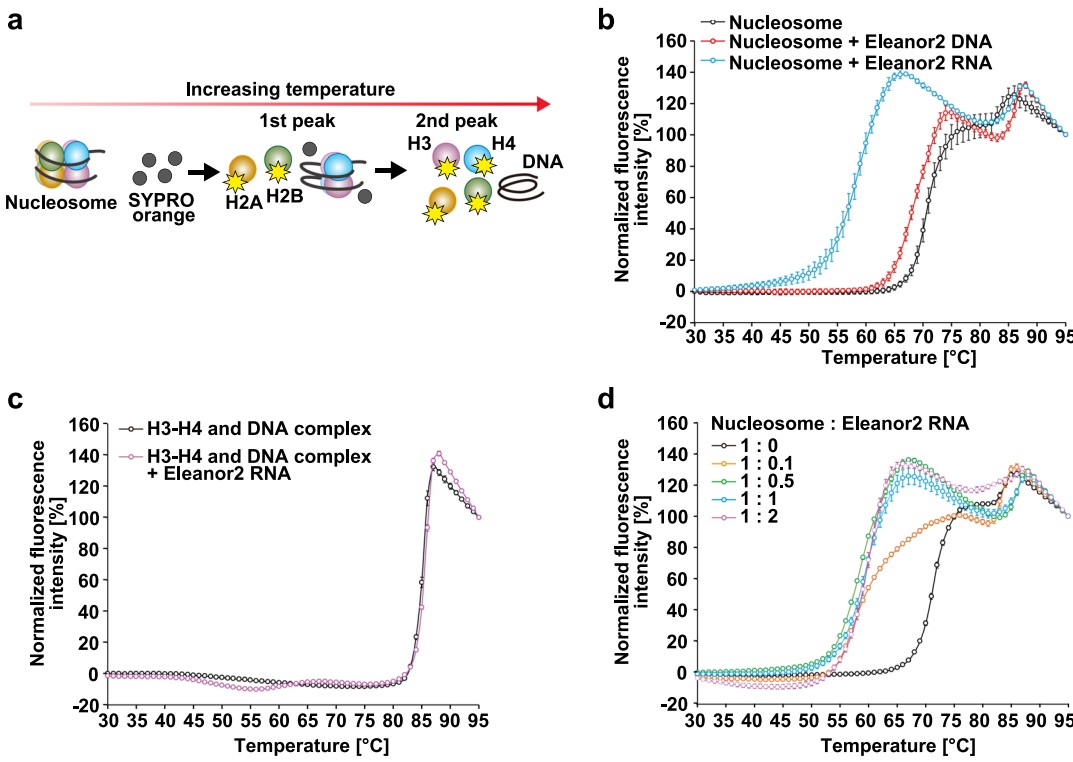

**Fig. 2 The Eleanor2 RNA destabilizes the nucleosome. a** Schematic representation of the nucleosome thermal stability assay. In this assay, SYPRO Orange binds to the hydrophobic surfaces of heat-denatured histones, but not DNA and RNA. The fluorescence signal from SYPRO Orange bound to histones is detected. The first and second peaks correspond to the dissociation phases of the H2A–H2B and H3–H4 complexes from the nucleosome, respectively. **b** Thermal stability curves of the nucleosome (1.25 μM) in the presence or absence of Eleanor2 DNA (0.62 μM) or RNA (1.25 μM). The fluorescence intensity was plotted against the temperature (from 30 °C to 95 °C). The means ± s.d. (*n* = 3) are shown. **c** Thermal stability curves of the H3–H4 and DNA complex (1.25 μM) in the presence or absence of the Eleanor2 RNA (1.25 μM). The means ± s.d. (*n* = 3) are shown. **d** Thermal stability curves of the nucleosome (1.25 μM) with increasing amounts of the Eleanor2 RNA (0.125, 0.625, 1.25, and 2.5 μM). Means ± s.d. (*n* = 3) are shown. The source data for the thermal stability assay are shown in Supplementary Fig. 6.

destabilization is not dependent on the DNA sequence of the nucleosome.

**The H2A–H2B eviction activity is dependent on the RNA length.** We next tested the nucleosome destabilizing activities of short fragments of Eleanor2. The secondary structure of the Eleanor2 RNA was predicted (Fig. 3a), and the Eleanor2 (250–302) and Eleanor2 (320–447) RNA fragments were prepared (Fig. 3b). We selected these regions, because they are predicted to form the secondary structural domains in the sequences highly conserved among mammals. These Eleanor2 fragments exhibited the H2A–H2B eviction activity (Fig. 3c).

Interestingly, we found that the 656 nt RNA (Eleanor2 FL) exhibited higher H2A–H2B eviction activity than the 128 nt RNA (Eleanor2 320–447), and the activity of the 53 nt RNA (Eleanor2 250–302) was the lowest among them (Fig. 3c). These results indicated the RNA-length dependency for the H2A–H2B eviction activity.

**The H2A–H2B eviction activity is shared among RNAs.** We investigated whether other nuclear ncRNAs, MALAT1, DSCAM-AS1, and XIST, have the same function as the Eleanor2 RNA. The MALAT1 ncRNA (~8000 nucleotides) was originally found as a metastasis-associated lung adenocarcinoma transcript 1[34], and it

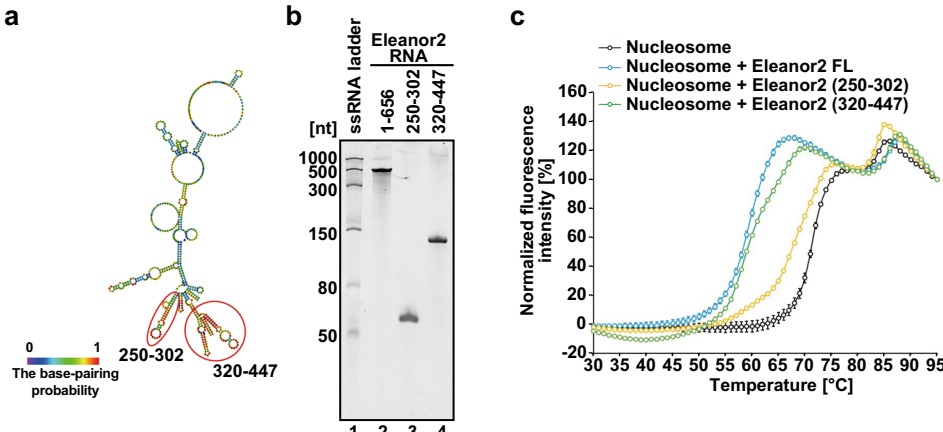

**Fig. 3 Thermal stability assay with the Eleanor2 RNA fragments. a** The Eleanor2 RNA secondary structures predicted by CentroidFold. RNA fragments employed in this assay are encircled by red ellipses with base numbers, 250–302 and 320–447. **b** Purified Eleanor2 RNA fragments transcribed in vitro were analyzed by 8% polyacrylamide/7 M urea denaturing gel electrophoresis with ethidium bromide staining. The uncropped gel image is shown in Supplementary Fig. 5. **c** Thermal stability curves of the nucleosome (1.25 μM) with Eleanor2 RNA fragments. Equimolar amounts (in moles of nucleotides) of the full-length Eleanor2 RNA (1–656) (1.25 μM), Eleanor2 (250–302) (14.0 μM), and Eleanor2 (320–447) (6.17 μM) were added to each reaction solution. The fluorescence intensity was plotted against the temperature (from 30 °C to 95 °C). Means ± s.d. ($n = 3$) are shown. The source data for the thermal stability assay are shown in Supplementary Fig. 7.

is overexpressed in various kinds of cancers[35,36]. In general, MALAT1 is abundantly expressed and primarily localized to nuclear speckles, nuclear domains that are located in the proximity of the active site of transcription[35–38]. The breast cancer-specific DSCAM-AS1 is an ncRNA that is highly expressed in MCF7 cells, and exists in both the nucleus and cytoplasm[29]. The XIST ncRNA is the major component for X chromosome inactivation in mammalian female cells, in which genes are silenced[39]. To determine whether MALAT1, DSCAM-AS1, and XIST share the nucleosome destabilizing activity, we prepared a 3720-base RNA fragment containing the MALAT1 sequence, a 2463-base RNA fragment containing the full-length DSCAM-AS1 sequence, and a 3037-base RNA fragment containing the XIST sequence by in vitro transcription (Supplementary Fig. 2a–c, and Supplementary Data 2 for RNA sequences). The MALAT1 sequence used in this study contains a region responsible for speckle targeting[27,40]. We then performed a thermal stability assay. As shown in Fig. 4a, b, and c, similar to the Eleanor2 RNA, the MALAT1, DSCAM-AS1, and XIST RNAs drastically weakened the H2A–H2B association with the nucleosome. Our thermal stability assay also revealed that the ESR1 mRNA, which is mainly located in the cytoplasm, destabilized the nucleosome in vitro (Fig. 4d; Supplementary Fig. 2d). Therefore, the H2A–H2B eviction activity may not be a consequence of a specific RNA sequence. In contrast, we found that poly(U), which may not contain obvious secondary structure[41], did not destabilize the nucleosome (Fig. 4e; Supplementary Fig. 2e), like the Eleanor2 DNA (Fig. 2b; Supplementary Fig. 3). This suggested that the RNA-mediated H2A–H2B eviction may require unbiased ribonucleotide sequences and/or RNA secondary structures.

**Eleanor2 promotes nucleosome depletion in LTED cells**. We finally investigated whether the Eleanor2 RNA contributes to the nucleosome depletion in vivo. We asked whether it is required for the high chromatin accessibility of the *ESR1* upstream region in the LTED cells, as shown in Fig. 1. To do so, we knocked down the Eleanor2 RNA in the LTED cells (Eleanor2 LNA), using the antisense LNA (locked nucleic acid) GapmeR (Fig. 5a). As expected, the FISH analysis showed that the Eleanor clouds disappeared with the knockdown (Eleanor2 LNA), while they remained intact in the control cells (Control LNA) (Fig. 5b). We

then performed ATAC-seq with two replications, and FAIRE-qPCR to investigate the changes in the chromatin accessibility upon the Eleanor2 knockdown (Fig. 5c–e; Supplementary Fig. 4a, b). Both analyses revealed that the chromatin accessibility in the *ESR1* upstream region (A–C sites) was reduced in the Eleanor2-knockdown cells, as compared with the control cells (Fig. 5c–e). In contrast, the chromatin accessibility at the D site and in the *ERBB2* region, which are located farther away from the *Eleanor2*-coding region, was not affected by the Eleanor2 depletion (Fig. 5c–e; Supplementary Fig. 4b). These results indicated that Eleanor2 contributes to the local chromatin accessibility near the site of its transcription and may destabilize nucleosomes in vivo.

## Discussion

The existence of ncRNAs has been detected, mainly by transcriptome analyses[30]. Many ncRNAs are localized in the nucleus and expressed in a tissue- or disease-specific manner, and thus they are considered to be important regulatory factors for gene expression, development, and disease[6,15,18,42]. Although various mechanisms have been proposed for ncRNAs[7–13], their involvement in nucleosome dynamics has not been demonstrated. In this study, we showed that a nuclear ncRNA, Eleanor2, drastically destabilized nucleosomes in vitro and in vivo.

Eleanors are produced in the LTED cells derived from breast cancer cell lines, and specifically accumulate around the *ESR1* gene, where they may stimulate its transcription. Eleanors are required for LTED cells to proliferate in the absence of estrogen, which recapitulates the endocrine therapy resistance of breast cancer. Suppression of Eleanor RNAs reduces the ESR1 transcription. Therefore, Eleanors may be a good therapeutic target for breast cancers[25,43,44]. However, the mechanism by which Eleanor RNAs stimulate the ESR1 transcription has remained enigmatic. Interestingly, our FAIRE-seq data analysis, ATAC-seq, and FAIRE-qPCR results suggested that, in the LTED cells, nucleosomes are destabilized at the Eleanor-accumulating sites around the *ESR1* gene (Figs. 1, 5). Consistently, we found that the Eleanor2 RNA, one of the highly produced Eleanors, destabilizes the nucleosome by evicting the H2A–H2B dimers from the nucleosome in vitro (Figs. 2, 3). These results provide evidence that an ncRNA can directly modulate the chromatin structure. A comprehensive analysis of

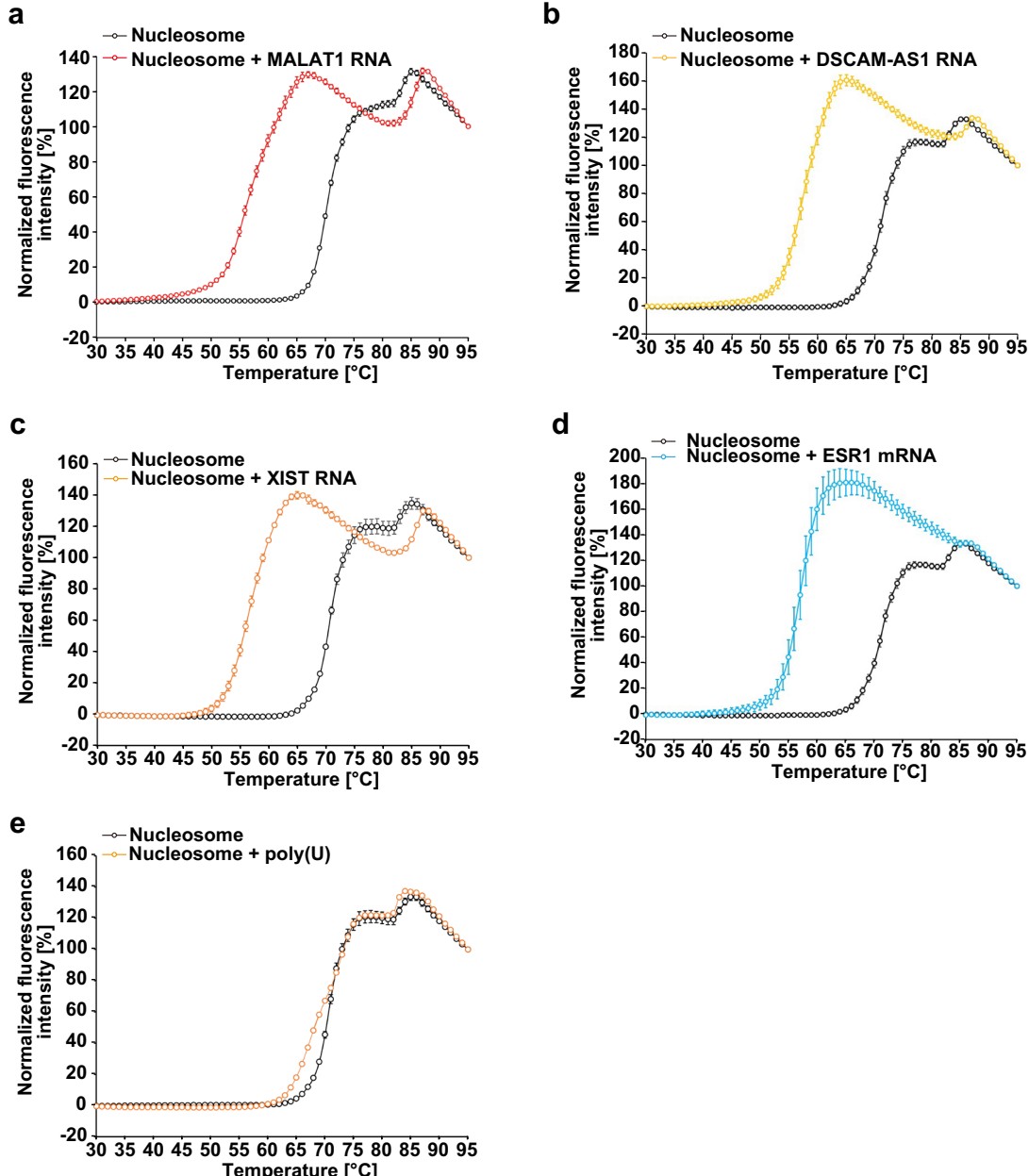

**Fig. 4 Thermal stability assay with other ncRNAs. a** Thermal stability curves of the nucleosome (1.25 μM) in the presence of MALAT1 RNA (0.22 μM). The amount of MALAT1 RNA (0.22 μM) corresponds to 1.25 μM Eleanor2 RNA in weight. Means ± s.d. (*n* = 3) are shown. **b** Thermal stability curves of the nucleosome (1.25 μM) in the presence of DSCAM-AS1 RNA (0.33 μM). The amount of DSCAM-AS1 RNA (0.33 μM) corresponds to 1.25 μM Eleanor2 RNA in weight. Means ± s.d. (*n* = 3) are shown. **c** Thermal stability curves of the nucleosome (1.25 μM) in the presence of XIST RNA (0.27 μM). The amount of XIST RNA (0.27 μM) corresponds to 1.25 μM Eleanor2 RNA in weight. Means ± s.d. (*n* = 3) are shown. **d** Thermal stability curves of the nucleosome (1.25 μM) in the presence of ESR1 mRNA (0.46 μM). The amount of ESR1 mRNA (0.46 μM) corresponds to 1.25 μM Eleanor2 RNA in weight. Means ± s.d. (*n* = 3) are shown. **e** Thermal stability curves of the nucleosome (1.25 μM) in the presence of poly(U) (265 ng/μl). The amount of poly(U) (265 ng/μl) corresponds to 1.25 μM Eleanor2 RNA in weight. Means ± s.d. (*n* = 3) are shown. The source data for the thermal stability assay are shown in Supplementary Fig. 8.

chromatin-interacting RNAs has been performed by the GRID-seq method, which captures global RNA interactions with genomic DNA[45]. Intriguingly, large sets of ncRNAs are localized in *cis*, and are especially enriched in active promoters and enhancers[46,47]. These characteristics are also conserved in the Eleanor RNAs[25]. Therefore, ncRNA species, which are localized around transcription regulation sites, directly modulate the chromatin structure and destabilize the nucleosomes suppressing gene expression.

We would like to stress that the Eleanor2 transcript is required to disassemble nucleosomes and create an open chromatin state. Our ATAC-seq and FAIRE-qPCR with the Eleanor2 knockdown showed that the Eleanor2 depletion reduced the nucleosome-free DNA in the Eleanor2 transcribed and neighboring regions (Fig. 5c–e). Our in vitro experiments revealed that RNAs disassembled nucleosomes in the absence of transcription. However, our results do not exclude another important possibility that the transcriptional activity plays a crucial role in opening the

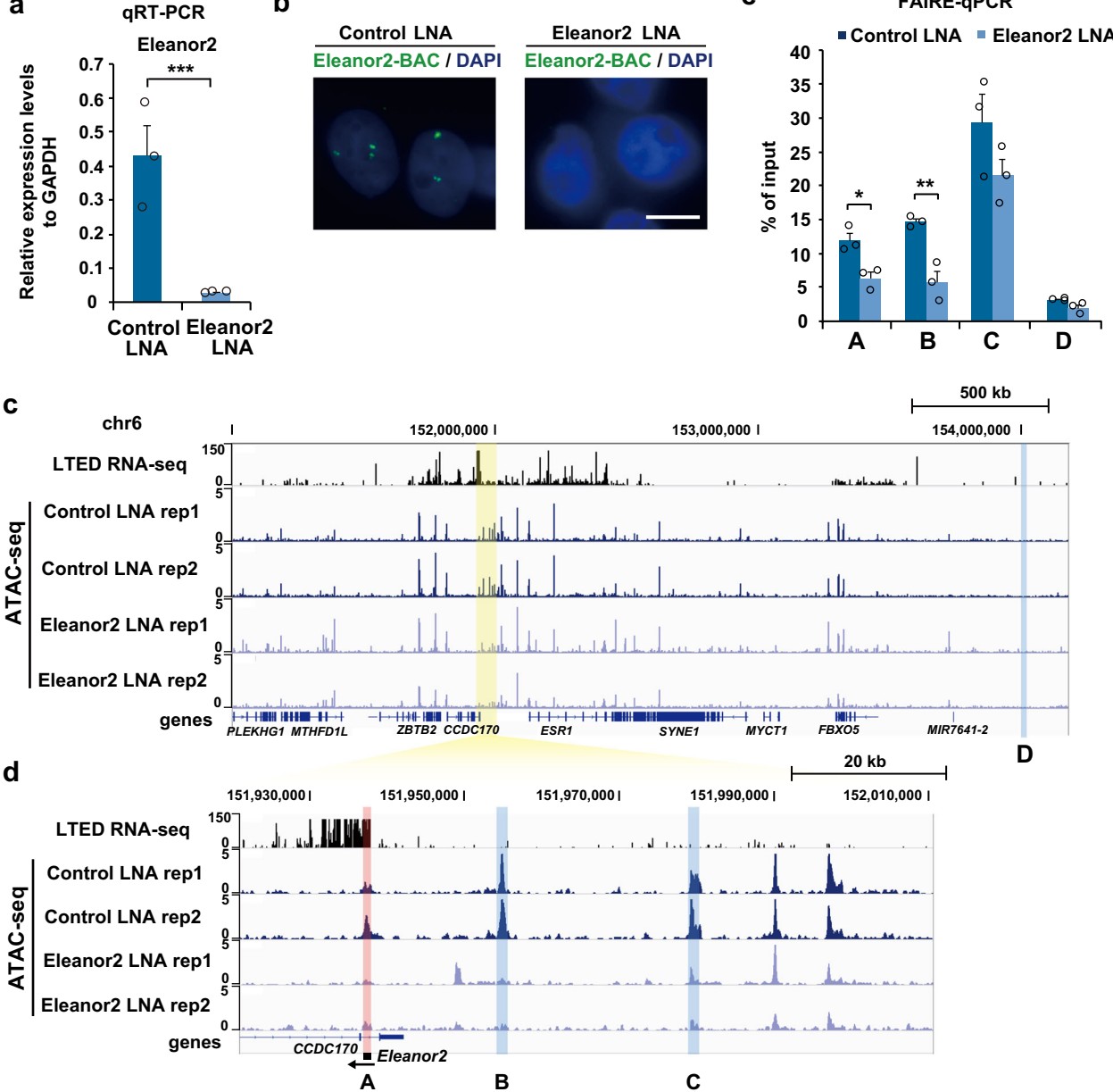

**Fig. 5 Eleanor2 RNA depletion reduced open chromatin conformation. a** Depletion of Eleanor2 RNA. LTED cells were treated with the indicated LNA (locked nucleic acid), and the expression of Eleanor2 RNA was measured by qRT-PCR. The values relative to the control GAPDH mRNA are shown. **b** The Eleanor RNA signal was suppressed with Eleanor2 RNA depletion. The Eleanor2-BAC was used as the probe for RNA FISH (green). Scale bar, 10 μm. **c** ATAC-seq showing that chromatin accessibility was reduced with Eleanor2 RNA depletion in LTED cells. Overview of the region, including the *Eleanor2* site (highlighted in yellow). The RNA-seq[25] and ATAC-seq tracks of LTED cells with the control LNA and Eleanor2 LNA were aligned against the genome reference GRCh37/hg19. **d** Enlarged view of the *Eleanor2* site. ATAC-seq peaks of A, B, and C were reduced by the Eleanor2 depletion. **e** FAIRE-qPCR revealed that nucleosome-depleted DNA was decreased with Eleanor2 RNA depletion in LTED cells. Values represent amounts of DNA in the nucleosome-free fraction relative to the input DNA. Data presented in **a**, **e** are means ± s.e.m. (*n* = 3 biologically independent samples). *P*-values were calculated using the unpaired, two-tailed, Student's *t* test (*$P < 0.05$, **$P < 0.01$, ***$P < 0.001$).

chromatin near the *Eleanor2*-coding region. It is plausible that because there is more transcription in this region, there is more open chromatin.

The nucleosome destabilization activity was detected in both the Eleanor2 and ESR1 mRNAs in vitro (Figs. 2, 4). On the other hand, only the *Eleanor2*-coding and neighboring regions showed high chromatin accessibility in vivo, and the *ESR1*-coding region did not (Fig. 1a). This discrepancy may occur because more Eleanor2 RNA stays at its own transcription site, creates the

Eleanor RNA cloud, and increases the local concentration of the RNAs. It therefore mimics the in vitro system, where the RNAs are abundantly provided to the nucleosome. The ESR1 mRNA, on the other hand, does not create the cloud[25], and efficiently moves away from the nucleus to the cytoplasm for protein translation in cells. The low concentration of the ESR1 mRNA around the chromatin may be incapable of destabilizing the nucleosome in vivo, even though the RNA itself has this ability, as shown in vitro. We consider the local RNA-concentrating activity and/or

the RNA-exporting activity to be additional layers that regulate the nucleosome destabilization by RNAs in cells.

The facultative heterochromatin component, PRC2, is a well-studied RNA-binding and chromatin-associating protein complex[48–50]. PRC2 binds to ncRNA such as the XIST RNA, and may be recruited to specific sites in the genome[51–54]. The heterochromatin mark, H3K27 trimethylation, is then introduced by PRC2 in chromatin around its recruited sites. In this case, the ncRNA functions as a guide for directing PRC2 to the specific sites, and may not directly affect the chromatin structure. Consistent with this idea, the ncRNA binding competes with the nucleosome binding to PRC2[55]. In contrast to the ncRNAs bound to PRC2, we found that the Eleanor2 RNA did not stably bind to the nucleosome, but instead directly modulated the nucleosome structure (Figs. 1–3, 5).

In this study, we found that the RNA-mediated H2A–H2B eviction may be a common feature for RNAs with unbiased sequences. Our thermal stability assay showed that a fragment of XIST, which forms inactive chromatin, also destabilizes the nucleosome (Fig. 4c). Therefore, in the inactive X chromosome, the H2A–H2B eviction activity of the RNA may be suppressed by factors that bind to the XIST RNA, or the direct RNA–nucleosome contact may be hindered by chromatin-binding or RNA-binding proteins. Further studies will be required to solve this interesting issue.

The H2A–H2B eviction activity is conserved among nuclear ncRNAs. RNA may directly bind to H2A–H2B and remove it from the nucleosome. The H2A–H2B depletion makes the nucleosome unstable, and the resulting nucleosome without an H2A–H2B dimer may become a target for nucleosome remodelers and/or histone chaperones, which are required for creating open chromatin regions by nucleosome depletion. Nuclear RNAs may also function as a factor to recruit nucleosome remodelers[56].

Many ncRNAs are highly expressed in cancer cells, and may function in transcription regulation[7]. This could occur through modulation of the chromatin architecture, similar to the Eleanor2 RNA. Therefore, the H2A–H2B eviction activity exhibited by the Eleanor2 RNA may be a common feature in a certain group of chromatin-associated ncRNAs. In addition, histone modifications are also perturbed in numerous cancer cells[57]. In this study, we performed in vitro assays with the nucleosomes without histone modifications. Some histone modifications reportedly affect the nucleosome stability[58,59]. It is intriguing to study the relationship between histone modifications and RNA-mediated nucleosome destabilization.

## Methods

**Cell culture**. The human immortalized normal breast cell line, MCF10A (ATCC), was cultured in DMEM/F12 with 5% horse serum (Gibco), 0.5 μg/ml hydrocortisone (Sigma), 100 ng/ml cholera toxin (Bio Academia), 10 μg/ml insulin (Sigma), and 20 ng/ml epidermal growth factor (PeproTech). The human estrogen receptor-positive breast cancer cell line, MCF7 (ATCC), was grown in RPMI 1640 containing 10% FBS (Corning). LTED cells were established by culturing MCF7 cells in phenol red-free RPMI 1640 (Wako), containing 4% dextran-coated charcoal-stripped FBS (Thermo Fisher Science), for 3–8 months. The cells were cultured at 37 °C with a humidified 5% $CO_2$ atmosphere.

**Analyses of the FAIRE-seq and RNA-seq data**. The FAIRE-seq data of MCF10A (GSM2175778) and LTED (GSM925735) cells[24,28] were obtained from the NCBI Gene Expression Omnibus (GEO) database. The data were aligned using the Burrows-Wheeler Aligner (BWA) (v. 0.7.12)[60] to the human genome (build hg19, GRCh37), and PCR duplicates were removed using MarkDuplicates (Picard v. 1.136 http://broadinstitute.github.io/picard/). We created TDF files normalized with read/million, using the Integrative Genomics Viewer (IGV v. 2.3.68)[61,62] tools with parameters set to count -z 5 -w 25 -e 200. The peak signals were visualized using IGV. The RNA-seq data of LTED cells were from our previously published data (DRA001006)[25].

**qRT-PCR**. The total RNA was isolated from cultured cells with TRIzol (Invitrogen). Reverse transcription was performed with ReverTra Ace qPCR RT Master Mix with gDNA Remover (TOYOBO). qRT-PCR was performed with SYBR green fluorescence on a Step One Plus system (Applied Biosystems). Values were normalized to GAPDH expression before calculating the relative expression levels. Primer sets used for analysis are listed in Supplementary Table 1.

**RNA fluorescence in situ hybridization (FISH)**. RNA-FISH was performed as follows. Briefly, cells grown on coverslips were fixed with 4% PFA + 0.5% Triton X-100, and permeabilized with 0.5% Saponin/0.5% Triton X-100/PBS. The cells were incubated with a hybridization mixture containing 5–10 μg/ml probe DNA, at 37 °C for 48 h. The BAC-DNA probe (RP11-66L11 (Eleanor2-BAC)) was labeled with digoxigenin or biotin using a nick translation mixture (Roche), according to the manufacturer's protocol. After hybridization, the coverslips were washed three times with 2 × SSC/50% formamide, and three times with 2 × SSC at 37 °C for 5 min each, and signals were detected with FITC-anti-digoxigenin (Roche), rhodamine-anti-digoxigenin (Roche), Alexa Fluor 488-anti-streptavidin (Molecular Probes), or Cy3-streptavidin (Jackson ImmunoResearch). DNA was counterstained with 4′,6-diamidino-2-phenylindole (DAPI) or Hoechst.

For dual-color RNA-FISH, BAC-DNA probes, RP11-66L11 (Eleanor2-BAC), RP11-450E24 (ESR1-BAC), and RP1-63I5 (ESR1-BAC2) and RP11-94L15 (ERBB2-BAC)) were labeled with Cy3-dUTP (Enzo) or Green 496 dUTP (Enzo) by a nick translation mixture (Abbott), according to the manufacturer's instructions. The FISH procedure is the same as that described above, except for the following wash step. After hybridization, the coverslips were washed twice with 2 × SSC/50% formamide, and three times with 2 × SSC/0.05% Tween 20 at 42 °C for 5 min.

FISH images were obtained with an IX-71 microscope or IX-83 microscope (Olympus) equipped with a 60 × NA1.0 Plan Apo objective lens or 60 × NA1.35 Universal Plan Super Apochromat oil immersion objective lens (Olympus), a cooled CCD camera (Hamamatsu), and image acquisition software (Lumina Vision, Mitani Corporation or cellSens, Olympus). For comparisons, corresponding images were obtained under identical image-capturing conditions.

**FAIRE-qPCR**. FAIRE-qPCR was performed as follows. MCF10A and LTED cells (~$3 × 10^6$ cells) were cross-linked with 1% formaldehyde for 5 min at room temperature. Cells were washed twice with PBS, and then the supernatant was removed by aspiration and the cell pellet was flash-frozen in liquid nitrogen. The frozen and cross-linked cells were stored at −80 °C. After adjusting the cell number to $3 × 10^6$, nuclear extraction was performed with lysis buffer, and DNA was fragmented using a Picoruptor (Nippon Gene) (10 cycles of 30 s ON/30 s OFF). Ten percent input was stored from the fragmented DNA, and the supernatant (free chromatin) was collected after two extractions with phenol–chloroform–isoamyl alcohol. Cross-links were reversed for 4 h at 65 °C with an RNase cocktail. The input DNA fraction was treated with Proteinase K at 50 °C for 1 h, and purified with a QIA-quick PCR Purification Kit (Qiagen). DNA enrichment in the FAIRE samples was determined using a qPCR analysis with a Step One Plus system (Applied Biosystems) and SYBR green fluorescence. The cycle number required to reach the threshold was recorded and analyzed. PCR was performed using the supernatant DNA (FAIRE sample) and the input DNA (control sample). Values were normalized to the input DNA, which was used to make a standard curve. Primer sets used for analysis are listed in Supplementary Table 2.

**Histone purification**. Human histones (H2A, H2B, H3.1, and H4) were expressed in Escherichia coli cells[63,64]. Briefly, His$_6$-tagged histones were produced and recovered in the insoluble fraction, and were denatured with 50 mM Tris-HCl buffer (pH 8.0), containing 7 M guanidine hydrochloride, 500 mM NaCl, and 5% glycerol. The His$_6$-tagged histones were then purified by Ni-NTA agarose chromatography (Qiagen) under denaturing conditions. The His$_6$-tag peptide was cleaved with thrombin protease. The histones without the His$_6$-tag peptide were purified by Mono S column chromatography (GE Healthcare). The purified histones were dialyzed against water and freeze-dried.

**Nucleosome purification**. The H2A–H2B and H3.1–H4 complexes were reconstituted as follows[65]. Lyophilized H2A and H2B were mixed at a 1:1 molar ratio and dissolved in denaturing buffer (20 mM Tris-HCl (pH 7.5), 20 mM 2-mercaptoethanol, and 7 M guanidine hydrochloride). Lyophilized H3.1 and H4 were also mixed at a 1:1 molar ratio and dissolved in the same buffer. The mixtures were dialyzed against refolding buffer (10 mM Tris-HCl (pH 7.5), 5 mM 2-mercaptoethanol, 1 mM EDTA, and 2 M NaCl), and the H2A–H2B and H3.1–H4 complexes were refolded. The resulting complexes were purified by chromatography on a HiLoad16/60 Superdex 200 size-exclusion column (GE Healthcare), equilibrated with refolding buffer. For nucleosome reconstitution, the histone complexes (H2A–H2B and H3.1–H4) were mixed with a purified DNA fragment (the palindromic 146 base-pair alpha-satellite DNA or the 147 base-pair MMTV-A DNA). Nucleosomes were reconstituted by the salt-dialysis method, and were separately purified by native polyacrylamide gel electrophoresis using a Prep Cell apparatus (Bio-Rad)[31], under conditions with 20 mM Tris-HCl (pH 7.5) and 1 mM dithiothreitol.

**Preparation of RNA fragments**. The Eleanor2 RNAs (full length and fragments), MALAT1 RNA, and DSCAM-AS1 RNA were transcribed in vitro. The DNA fragments encoding the Eleanor2 RNAs, the MALAT1 RNA, and the ESR1 mRNA containing an EcoRV site or the DSCAM-AS1 RNA and the XIST RNA containing an NruI site at the 3′-end were each inserted into the pGEM T-easy vector (Promega) downstream of the T7 promoter sequence. The plasmids containing the *Eleanor2* and *MALAT1* sequences were linearized by digestion with *Eco*RV. The plasmids containing the *DSCAM-AS1* and *XIST* sequences were linearized by digestion with *Nru*I. These linearized products were used as templates for in vitro transcription. RNAs were produced using a RiboMAX T7 Express kit (Promega). The resulting RNAs contained GGG (derived from the T7 promoter) at the 5′-terminus and GAC (derived from digestion with *Eco*RV) or UCG (derived from digestion with *Nru*I) at the 3′-terminus. The products were purified by phenol/chloroform treatment followed by ethanol precipitation, or by MicroSpin G-25 Columns (GE Healthcare). Poly(U) (product number: P9528) was purchased from Sigma-Aldrich. For the thermal stability assay, the samples were dialyzed against water. The 662 base-pair double-stranded DNA fragment encoding the Eleanor2 sequence was amplified by PCR, using the following primers: forward: 5′-GGGCAGGAAGCTGTGCTGTGCTCACCAGACACT-3′ and reverse: 5′-ATC CGCCTCTGAAGTTTGCATCATCATAGAGCT-3′. The resulting DNA fragment was purified by native polyacrylamide gel electrophoresis using a Prep Cell apparatus (Bio-Rad), and dialyzed against water.

**Thermal stability assay**. The thermal stability assay was performed as follows[32]. The nucleosome or the complex of H3–H4 and DNA containing the palindromic alpha-satellite 146 base-pair DNA was tested in the presence or absence of RNA in 19 µl of 12.6 mM Tris-HCl (pH 7.5) buffer, containing 5 × SYPRO Orange (Sigma-Aldrich), 100 mM NaCl, and 0.6 mM dithiothreitol, with a temperature gradient from 26 to 95 °C in steps of 1 °C/min. The fluorescence signals of the SYPRO Orange were measured at each temperature by a StepOnePlus Real-Time PCR System (Applied Biosystems).

**RNA secondary structure prediction**. We employed CentroidFold[66] to predict the secondary structure of a given RNA sequence, using the following command line:
centroid_fold -e McCaskill -g 4 input.seq which performs the RNA secondary structure predictio n based on the Turner's energy model in combination with the γ-centroid estimator of γ = 4. In addition, the base-pairing probability matrix $\{p_{ij}\}_{i<j}$ ($p_{ij}$ is the probability that the $i$th and $j$th positions form a base pair)[67] was also computed by CentroidFold.

**Determination of RNA secondary structure domains**. An RNA sequence was divided into a set of 10 nt blocks, and every combination of successive blocks is a domain candidate. For a domain candidate, $D$ and $D^c$ denote a set of nucleotides inside and outside the domain, respectively. We then define two normalized sums of base-pairing probabilities as follows:

$$p^{(i)}(D) := \frac{\sum_{i,j \in D} p_{ij}}{|D|^2} \quad (1)$$

$$p^{(io)}(D) := \frac{\sum_{i,j \in (D^c \times D) \cup (D \times D^c)} p_{ij}}{(L - |D|)|D|} \quad (2)$$

where $p_{ij}$ is the base-pairing probability, and $L$ and $|D|$ are the lengths of an input sequence and a domain, respectively. In this study, we extracted a region $|D|$ that satisfies $p^{(i)}(D) > 0.003$ and $p^{(io)}(D) < 0.0003$, meaning that the region includes a stable local secondary structure.

**Transfection of cells with LNA**. Cells were transfected with antisense LNA (locked nucleic acid) GapmeR Negative Control A, Eleanor2 (synthesized and purchased from TAKARA) using RNAiMAX (Invitrogen). The cells were analyzed at 24 and 48 h after transfection. Target sequences of Antisense LNA are as follows. Control LNA (Negative Control A): 5′-AACACGTCTATACGC-3′, Eleanor2 LNA: 5′-TTTGGCCACATTGGAA-3′.

**Assay for Transposase-Accessible Chromatin using sequencing (ATAC-seq)**. ATAC-seq analyses were performed according to the published protocol[68]. Briefly, 50,000 cells were collected, and nuclei were prepared. The ATAC reactions with Transposase (Illumina) were performed at 37 °C for 30 min. The transposed fragments were purified with a DNA Clean & Concentrator-5 Kit (ZYMO RESEARCH), amplified by PCR using NEBNext High-Fidelity 2× PCR Master Mix (NEB) with sequence primers, and sequenced by Mi-seq (Illumina). The sequenced data were aligned with the Bowtie2 program (v. 2.3.4.3)[69] to the human genome (build hg19, GRCh37), and PCR duplicates were removed using MarkDuplicates (Picard v. 1.136 http://broadinstitute.github.io/picard/). We made TDF files normalized with read/million, using the Integrative Genomics Viewer (IGV v. 2.3.68)[61,62] tools with parameters set to count -z 5 -w 25 -e 150. The peak signals were visualized using IGV. Pearson's correlation coefficients were calculated by the BamCompare module of the deeptools program[70], with a bin size of 150 bp.

**Statistics and reproducibility**. The data in Figs. 1c, f, 5a, e are mean ± s.e.m. of three biologically independent experiments. *P*-values were assessed using the unpaired and two-tailed Student's *t* test (Figs. 1c, f, 5a, e), with MS Excel. *$P < 0.05$, **$P < 0.01$, ***$P < 0.001$. $P < 0.05$ was considered statistically significant. Coverage was adopted as reproducibility metrics for ATAC-seq biological replicates (fraction of reads/million uniquely mappable reads), and the correlation was calculated using Pearson's correlation coefficient with the deeptools program[70]. The data in Figs. 2b–d, 3c, and 4a–e were analyzed with MS Excel, and are presented as mean ± s.d. *n* represents the number of independent experiments.

**Reporting summary**. Further information on research design is available in the Nature Research Reporting Summary linked to this article.

## Data availability
ATAC-seq raw data are available in the DDBJ Sequenced Read Archive, under the accession number DRA008967. All other data are available in the Supplementary Information or from the authors upon reasonable request.

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

## Acknowledgements
We are grateful to Ms. Yukari Iikura (The University of Tokyo) for her assistance. This work was supported in part by JSPS KAKENHI grant numbers JP17H01408 [to H.K.], JP18H05534 [to H.K.], JP18H05531 [to N.S.], JP16H04744 [to N.S.], JP18K19479 [to M. N.], JP17K15043 [to Y.A.], JP19K23714 [to R.F.], JP19K23736 [to T.Y.], and JP19K21184 [to Y.I.]. This work was partly supported by JST CREST grant number JPMJCR16G1 [to H.K.], JST ERATO grant number JPMJER1901 [to H.K.], and also by the Platform Project for Supporting Drug Discovery and Life Science Research (Basis for Supporting Innovative Drug Discovery and Life Science Research (BINDS)) from AMED under Grant Number JP19am0101076 [to H.K.], and by grants from the Takeda Science Foundation [to N.S] and AMED CREST grant number JP16gm0510007 [to M.N.], and Mitsubishi Foundation [to M.N.]. R.F. was supported by a JSPS Research Fellowship for Young Scientists, JP16J10043. Funding for open access charge: The University of Tokyo.

## Author contributions
R.F. and Y.A. performed biochemical analyses. T.Y., S.F., H.T., Y.I., Y.S., L.Y., R.M. and M.N. performed genome analyses and cell biological analyses. M.H. analyzed and predicted the secondary structure of Eleanor2. H.K. and N.S. conceived, designed, and supervised all of the work, and H.K. wrote the paper. All of the authors discussed the results and commented on the paper.

## Competing interests
The authors declare no competing interests.
