## [Peer Review File · Communications Biology]

Reviewers' comments:

Reviewer #1 (Remarks to the Author):

In this manuscript, Fujita and colleagues investigate the effect of non-coding RNAs produced from the estrogen receptor-alpha locus on local nucleosomal stability. They observe that in breast cancer cells, the expression of the lncRNA Eleanor2 correlates with local increased chromatin accessibility, while, in vitro, the purified RNA destabilizes nucleosomes. They observe the same property for MALAT1 and DSCAM-AS1, proposing that this activity may be key for the role of lncRNAs in the activation of gene expression.

This is an interesting and clearly written paper. However, while it proposes an interesting idea, the data presented don't definitively support it. Some additional experiments should be performed to more robustly demonstrate the model proposed.

First, the claim that Eleanor2 induces the nucleosomal repositioning is solely based on correlation between expression and FAIRE data. Transcription and chromatin factors are recruited at the locus and also may be involved in the nucleosomal repositioning. In order to show that the RNA is key for it, the authors should perform Eleanor2 knockdown in LTED cells and analyze the effect on nucleosomal density.

Second, the authors propose that the property observed in vitro for Eleanor2 is characteristic of lncRNAs. However, it might be a property of RNA in general. To test this, also protein-coding mRNAs and additional unrelated RNA sequences should be tested in this assay. It would be interesting to see also the effect of lncRNAs known to mediate gene silencing, such as XIST.

In any case, the authors should discuss how they envision that noncoding RNAs are able to regulate nucleosomal positioning in vivo.

Reviewer #2 (Remarks to the Author):

In the presented study, Kurumizaka and colleagues perform an analysis to dissect the role of Eleanor2 non-coding RNA on nucleosome stability in vitro and in cells. They interestingly find that ncRNA destabilizes nucleosomes by inducing H2A-H2B eviction in vitro and depletion of nucleosomes in cells. These findings are very interesting however there are a few important controls missing and a few points that need clarification:

1. The authors use alpha satellite DNA for their in vitro analyses, but this is a strong nucleosome positioning sequence with no physiological reference. It is difficult to reconstitute nucleosomes with the actual native sequence but it is important to test other types of nucleosome positioning sequences as they might have a different sensitivity to the ncRNA. For example, the authors can test a sequence that has a weaker binding, such as MMTV, and see if the effect is similar or different.
2. The authors use the Eleanor2 DNA as a control for the in vitro analysis, showing no effect. But an additional important control is to use non-specific RNA with maybe different or no secondary structure.
3. The authors use recombinant histones that are unmodified. This has the advantage of a pure biophysical analysis, but is perhaps not physiological, as rarely histones are unmodified in vivo. It might be beyond the scope of this paper to test histones extracted from mammalian cells in their reconstituted assay, but at minimum the authors should refer to this issue in the discussion (as well as the implications of their results in that context).
4. The authors select Eleanor2 (250-302) and (320-447) based on the predicted structure, but don't

explain in the text why specifically they chose these structures for their testing as these are not the only predicted structures?

5. The authors implied several times that H2A-H2B eviction is a main cause for nucleosomal instability by ncRNA ("Therefore, the H2A-H2B eviction activity may be a common characteristic of chromatin-associated ncRNAs") but don't refer to this in the discussion (reasons or implications) or propose any mechanism for this effect (DNA unwrapping? Recruitment of remodelers? Direct competition with RNA?). The authors should add a section reviewing this to their discussion.

6. The authors find that "an Eleanor2 RNA (green signals in Fig. 4c) accumulated in the proximity of the other Eleanors." However, there is no negative control in this assay showing there is no colocalization with an unrelated gene/region.

7. The authors conclude that "Collectively, these data implied that the Eleanor2 RNA actually destabilizes the nucleosomes around the upstream region of the ESR1 gene in LTED cells" and that "we showed that a nuclear ncRNA, Eleanor2, drastically destabilized nucleosomes in vitro and in vivo." However, while the authors show direct destabilization in vitro, they don't show such link in cells and the data is correlative (localization and FAIRE-seq). To show a direct link in cells, the authors should show a knockdown or a rescue experiment. One example is to do RNAi to the Eleanor2 RNA showing stabilization of the same region by FAIRE-seq.

Minor points:

1. Introduction 3rd line H3-H4 tetramer (not dimers).
2. The authors mention that "...the Eleanor2-BAC probe (Fig. 1, green bar), which mainly detects the Eleanor2 RNA sequence." If it doesn't exclusively bind the Eleanor2-BAC probe, what else does it bind? Is that background in the system?

Reviewer #3 (Remarks to the Author):

This paper by Fujita et al look at how the noncoding RNA Eleanor2 affects nucleosome stability. Their data shows that in vitro, Eleanor2 is capable of disassembling H2A-H2B from the nucleosome but has no clear affect on H3-H4. They also show that this is length dependent, given that decreasing the size of Eleanor2 decreases its ability to disassemble the nucleosome. Finally, they show in LTED cells that regions in the genome corresponding to Eleanor2 are more in an open-chromatin state. The idea is that Eleanor2 upregulation in LTED cells leads to Eleanor2-induced disassembly of nucleosomes, however they do not investigate the idea any further (in cells).

The paper is well written and clear to follow; however, there are two clear problems with this paper. The first problem is that they do not show that the RNA used in vitro is folded properly. The second problem is they do not use any knockdown studies to show that Eleanor2 is directly responsible for the open chromatin state in LTED cells. At the very least, these caveats should be mentioned.

Below is more detailed criticism.

Methods: RNA folding is an issue. This issue pervades all the in vitro results in the paper.

Results

Fig 1

In LTED cells, the Eleanor region has a larger open chromatin state than MCF10A control cells. The idea is that because there is more Eleanor2 RNA, the Eleanor2 can disassemble nucleosomes, which leads to the open chromatin state. However, it is possible that because there is more transcription in this region, there is more open chromatin. The author should point this out.

Fig 2

A very quantitative way of showing Eleanor2 RNA helps remove H2A-H2B dimers from nucleosomes, and that it is concentration dependent. However, they do not show anything that suggests the RNA is folded correctly. Also it appears that DNA does affect H2A-H2B eviction, just not as much as the RNA did, and this should be discussed.

Fig 4

This shows that Eleanor2 is localized to the same regions as other Eleanor RNA's in the nucleus. It also confirms data from figure 1 which shows that regions in the Eleanor2 gene are more open chromatin. However, they should use RNA-knockdown studies to show that this is because the RNA destabilizes the nucleosome. Otherwise, it is possible that more transcription in the region leads to more open chromatin. The authors should point this out again to avoid misleading the readers.

Reviewer #1 (Remarks to the Author):

In this manuscript, Fujita and colleagues investigate the effect of non-coding RNAs produced from the estrogen receptor-alpha locus on local nucleosomal stability. They observe that in breast cancer cells, the expression the lncRNA Eleanor2 correlates with local increased chromatin accessibility, while, in vitro, the purified RNA destabilizes nucleosomes. They observe the same property for MALAT1 and DSCAM-AS1, proposing that this activity may be key for the role of lncRNAs in the activation of gene expression.

This is an interesting and clearly written paper. However, while it proposes an interesting idea, the data presented don't definitively support it. Some additional experiments should be performed to more robustly demonstrate the model proposed.

First, the claim that Eleanor2 induces the nucleosomal repositioning is solely based in correlation between expression and FAIRE data. Transcription and chromatin factors are recruited at the locus and also may be involved in the nucleosomal repositioning. In order to show that the RNA is key for it, the authors should perform Eleanor2 knockdown in LTED cells and analyze the effect in nucleosomal density.

We appreciate this critical comment. This is evidently important, as all three of the reviewers raised this point. Therefore, we performed new experiments to address this issue. We knocked down the *Eleanor2* RNA using Antisense LNA GapmeRs, and performed ATAC-seq and FAIRE-qPCR (new Fig. 5). First, we confirmed that the *Eleanor2* RNA expression and *Eleanor* RNA cloud were suppressed with the *Eleanor2*-knockdown in LTED cells (new Fig. 5a, b). We then found, by ATAC-seq and FAIRE-qPCR, that the chromatin accessibility at the *Eleanor2* locus and its neighboring region decreased (A, B, and C sites) with *Eleanor2* RNA depletion in LTED cells (new Fig. 5c, d). In contrast, other regions (D site and the *ERBB2* locus) that are farther away from the *Eleanor2*-coding region were not affected by the *Eleanor2* depletion (new Fig. 5c, d and Supplementary Figure 4b). We have described these new data in the text on page 7, line 29 to page 8, line 12.

Second, the authors propose that the property observed in vitro for Eleanor2 is characteristic of lncRNAs. However, it might be a property of RNA in general. To test this, also protein-coding mRNAs and additional unrelated RNA sequences should be tested in this assay. It would be interesting to see also the effect of lncRNAs known to mediate gene silencing, such as XIST.

According to this important comment, we have performed more thermal stability assays, as shown in the new Fig. 4. We found that the nucleosome destabilization ability is shared with the protein-coding mRNA and other lncRNAs, for both gene activation and silencing. The effect was RNA-length dependent. The nucleosome destabilization activity was not in either the DNA (Fig. 2) or unstructured-RNA (new Fig. 4). We describe these points in the text on page 7, lines 19-27.

In any case, the authors should discuss how do they envision that noncoding RNAs are able to regulate nucleosomal positioning in vivo.

Thank you for this comment. We created a new paragraph in the discussion, to describe how we envision the regulation of nucleosomal positioning by noncoding RNAs *in vivo*, on page 9, lines 18-30.

Reviewer #2 (Remarks to the Author):

In the presented study, Kurumizaka and colleagues perform an analysis to dissect the role of Eleanor2 non-coding RNA on nucleosome stability in vitro and in cells. They interestingly find that ncRNA destabilizes nucleosomes by inducing H2A-H2B eviction in vitro and depletion of nucleosomes in cells. These findings are very interesting however there are a few important controls missing and a few points that need clarification:

1. The authors use alpha satellite DNA for their in vitro analyses, but this is a strong nucleosome positioning sequence with no physiological reference. It is difficult to reconstitute nucleosomes with the actual native sequence but is important to test other types of nucleosome positioning sequences as they might have a different sensitivity to the ncRNA. For example, the authors can test a sequence that has a weaker binding, such as MMTV, and see if the effect is similar or different.

Thank you very much for this suggestion. In the revised manuscript, we performed a thermal stability assay in the presence of the nucleosomes reconstituted with the MMTV LTR sequence. We then confirmed that the RNA-mediated nucleosome destabilization occurs with the MMTV nucleosome, as well as the alpha-satellite nucleosome. These data are presented in the new Supplementary Figure 1f, and described in the second paragraph on page 6.

2. The authors use the Eleanor2 DNA as a control for the in vitro analysis, showing no effect.

But an additional important control is to use non-specific RNA with maybe different or no secondary structure.

Thanks again for this insightful comment. To test the RNA without obvious secondary structure, we performed a thermal stability assay with poly(U) RNA. We then found that the poly(U) RNA has minimal nucleosome destabilizing ability, as compared to the similar length of *Eleanor2* RNA. These new data are presented in Fig. 4e and Supplementary Figure 2e, and the results are described on page 7, lines 23-27.

3. The authors use recombinant histones that are unmodified. This has the advantage of a pure biophysical analysis, but is perhaps not physiological, as rarely histones are unmodified in vivo. It might be beyond the scope of this paper to test histones extracted from mammalian cells in their reconstituted assay, but at minimum the authors should refer to this issue in the discussion (as well as the implications of their results in that context).

Thank you very much. We agree that the effect of histone modifications on the nucleosome destabilization by RNA is an interesting topic. We plan to prepare modified histones at defined residues by chemical synthesis, which will be an excellent project for future studies. As suggested by this reviewer, we added the related discussion in the last paragraph of the discussion section in the revised manuscript, on page 10, lines 26-31.

4. The authors select Eleanor2 (250-302) and (320-447) based on the predicted structure, but don't explain in the text why specifically they chose these structures for their testing as these are not the only predicted structures?

We selected these structures for our testing, because they are predicted to form the secondary structural domains with sequences highly conserved among mammals. We describe this in the revised manuscript, on page 6, lines 25-26.

5. The authors implied several times that H2A-H2B eviction is a main cause for nucleosomal instability by ncRNA ("Therefore, the H2A-H2B eviction activity may be a common characteristic of chromatin-associated ncRNAs") but don't refer to this in the discussion (reasons or implications) or propose any mechanism for this effect (DNA unwrapping? Recruitment of remodelers? Direct competition with RNA?). The authors should add a section reviewing this to their discussion.

Thank you very much for this comment. In the revised manuscript, we refer to the proposed mechanism for the H2A-H2B eviction and its implications in the discussion section, on page 10, lines 7-14.

6. The authors find that “an Eleanor2 RNA (green signals in Fig. 4c) accumulated in the proximity of the other Eleanors.” However, there is no negative control in this assay showing there is no colocalization with an unrelated gene/region.

We apologize for not including a negative control in our previous FISH assay. We obtained new FISH data using a probe visualizing the *ERBB2* region. In the revised manuscript, we show that *Eleanors* and *ERBB2* are not colocalized in LTED cells (new Fig. 1d). *ERBB2* is encoded on chromosome 17 and activated in LTED cells (Tomita et al., Nat Commun, 2015). In addition, we have replaced the FISH images with new ones (*Eleanor2*-BAC/*ESR1*-BAC/Hoechst and *Eleanor2*-BAC/*ESR1*-BAC2/Hoechst) that were taken in the same experiment as the one in which the *Eleanor2* and *ERBB2* (*Eleanor2*-BAC/*ERBB2*-BAC/Hoechst) image was taken. We describe this result in the text, on page 4, line 29 to page 5, line 8.

7. The authors conclude that “Collectively, these data implied that the Eleanor2 RNA actually destabilizes the nucleosomes around the upstream region of the ESR1 gene in LTED cells” and that “we showed that a nuclear ncRNA, Eleanor2, drastically destabilized nucleosomes in vitro and in vivo.” However, while the authors show direct destabilization in vitro, they don’t show such link in cells and the data is correlative (localization and FAIRE-seq). To show a direct link in cells, the authors should show a knockdown or a rescue experiment. One example is to do RNAi to the Eleanor2 RNA showing stabilization of the same region by FAIRE-seq.

We again thank the reviewer for raising this important point. This is the same question as the one asked by reviewers #1 and #3. To answer it, we knocked down the *Eleanor2* RNA using Antisense LNA GapmeRs, and performed ATAC-seq and FAIRE-qPCR (new Fig. 5). First, we confirmed that the *Eleanor2* RNA expression and *Eleanor* RNA cloud were suppressed with the *Eleanor2*-knockdown in LTED cells (new Fig. 5a, b). We then found, by ATAC-seq and FAIRE-qPCR, that the chromatin accessibility at the *Eleanor2* locus and its neighboring region decreased (A, B, and C sites) with *Eleanor2* RNA depletion in LTED cells (new Fig. 5c, d). In contrast, other regions (D site and the *ERBB2* locus) that are farther away from the *Eleanor2*-coding region were not affected by the *Eleanor2* depletion (new Fig.

5c, d and Supplementary Figure 4b). We have described these new data in the text on page 7, line 29 to page 8, line 12.

Minor points:

1. Introduction 3rd line H3-H4 tetramer (not dimers).

We corrected the word, as suggested.

2. The authors mention that "...the Eleanor2-BAC probe (Fig. 1, green bar), which mainly detects the Eleanor2 RNA sequence." If it doesn't exclusively bind the Eleanor2-BAC probe, what else does it bind? Is that background in the system?

We thank the reviewer for raising this important point. *Eleanor2* is one of the *Eleanors*, a cluster of RNAs transcribed from approximately 0.7 Mb of the genomic region (Tomita et al., Nat Commun, 2015). *Eleanor2* is the most abundant RNA among *Eleanors*. Because of that, the *Eleanor2*-BAC probe mainly detects *Eleanor2*, and possibly some other minor *Eleanor* species. For simplicity, we deleted "mainly" from the sentence on page 4, line 31.

Reviewer #3 (Remarks to the Author):

This paper by Fujita et al look at how the noncoding RNA Eleanor2 affects nucleosome stability. Their data shows that in vitro, Eleanor2 is capable of disassembling H2A-H2B from the nucleosome but has no clear affect on H3-H4. They also show that this is length dependent, given that decreasing the size of Eleanor2 decreases its ability to disassemble the nucleosome. Finally, they show in LTED cells that regions in the genome corresponding to Eleanor2 are more in an open-chromatin state. The idea is that Eleanor2 upregulation in LTED cells leads to Eleanor2-induced disassembly of nucleosomes, however they do not investigate the idea any further (in cells).

The paper is well written and clear to follow; however, there are two clear problems with this paper. The first problem is that they do not show that the RNA used in vitro is folded properly. The second problem is they do not use any knockdown studies to show that Eleanor2 is directly responsible for the open chromatin state in LTED cells. At the very least, these caveats should be mentioned.

Below is more detailed criticism.

Methods: RNA folding is an issue. This issue pervades all the in vitro results in the paper.

Thank you very much. This is an important issue. To test the *Eleanor2* RNA folding *in vitro*, we analyzed the *Eleanor2* RNA and DNA in the two-gel system, using non-denaturing and denaturing PAGE. If the *Eleanor2* RNA is not folded, then the migration distances of the *Eleanor2* RNA and DNA would be the same in both non-denaturing and denaturing PAGE. If the *Eleanor2* RNA is folded, then it would migrate differently only in the non-denaturing PAGE, but not in the denaturing PAGE. Our experiments showed that the *Eleanor2* RNA migrated differently from the *Eleanor2* DNA only in the non-denaturing PAGE. Therefore, we consider that the *Eleanor2* RNA is folded *in vitro*. These new data are presented in the new Supplementary Figure 3, and explained in its figure legend.

Results

Fig 1

In LTED cells, the Eleanor region has a larger open chromatin state than MCF10A control cells. The idea is that because there is more Eleanor2 RNA, the Eleanor2 can disassemble nucleosomes, which leads to the open chromatin state. However, it is possible that because there is more transcription in this region, there is more open chromatin. The author should point this out.

Thank you for this important comment. As we describe later in detail, we added new experimental data. We knocked down the *Eleanor2* RNA using Antisense LNA GapmeRs, and performed ATAC-seq and FAIRE-qPCR (new Fig. 5). These new data showed that the *Eleanor2* depletion reduced the nucleosome-free DNA in the *Eleanor2* and neighboring regions. Therefore, we would like to mention that the *Eleanor2* transcripts are at least required to disassemble nucleosomes and create the open chromatin state.

However, we also agree with the reviewer that it is still possible that, because there is more transcription in this region, there is more open chromatin. In the new manuscript, we discuss this possibility on page 9, lines 9-17.

Fig 2

A very quantitative way of showing Eleanor2 RNA helps remove H2A-H2B dimers from nucleosomes, and that it is concentration dependent. However, they do not show anything that suggests the RNA is folded correctly. Also it appears that DNA does affect H2A-H2B eviction, just not as much as the RNA did, and this should be discussed.

As mentioned above, we tested whether the *Eleanor2* RNA folds *in vitro*. Our non-denaturing and denaturing gel systems revealed that the *Eleanor2* RNA is folded *in vitro*. These new data are presented in the new Supplementary Figure 3, and explained in its figure legend. In addition, we performed the thermal stability assay with poly(U), which may not form an obvious secondary structure. We then found that the poly(U) did not show a significant level of the H2A-H2B eviction activity (new Fig. 4e). In total, these results suggest that the *Eleanor2* RNA is folded *in vitro*, and the secondary structure of the RNA is important for the H2A-H2B eviction activity. This result, as well as the result with DNA, is described on page 7, lines 23-27. As the reviewer pointed out, minimal nucleosome destabilization with the *Eleanor2* DNA was observed in Fig. 2b. This may reflect nucleosome competition between the DNA assembled in the nucleosome and the newly added DNA, under the heated conditions.

Fig 4

This shows that Eleanor2 is localized to the same regions as other Eleanor RNA's in the nucleus. It also confirms data from figure 1 which shows that regions in the Eleanor2 gene are more open chromatin. However, they should use RNA-knockdown studies to show that this is because the RNA destabilizes the nucleosome. Otherwise, it is possible that more transcription in the region leads to more open chromatin. The authors should point this out again to avoid misleading the readers.

We again thank the reviewer for raising this important point. This is the same question as the one asked by reviewers #1 and #2. To answer this question, we added new experimental data. We knocked down the *Eleanor2* RNA using Antisense LNA GapmeRs, and performed ATAC-seq and FAIRE-qPCR (new Fig. 5). First, we confirmed that the *Eleanor2* RNA expression and *Eleanor* RNA cloud were suppressed with the *Eleanor2*-knockdown in LTED cells (new Fig. 5a, b). Then we found, by ATAC-seq and FAIRE-qPCR, that the chromatin accessibility at the *Eleanor2* locus and its neighboring region decreased (A, B, and C sites) with *Eleanor2* RNA depletion in LTED cells (new Fig. 5c, d). In contrast, other regions (D site and the *ERBB2* locus) that are farther away from the *Eleanor2*-coding region were not affected by the *Eleanor2* depletion (new Fig. 5c, d and Supplementary Figure 4b). We describe these new data in the text on page 7, line 29 to page 8, line 12.

REVIEWERS' COMMENTS:

Reviewer #1 (Remarks to the Author):

The authors have addressed all my comments.

Reviewer #2 (Remarks to the Author):

The authors have answered all of my questions and have provided additional explanations/clarifications where required. I am happy to accept this revised version as is and have no further remarks. Congratulations on a great story!

Reviewer #3 (Remarks to the Author):

we are happy with the revisions and recommend acceptance.